# Immune Responses Induced by Recombinant Membrane Proteins of *Mycoplasma agalactiae* in Goats

**DOI:** 10.3390/vaccines13070746

**Published:** 2025-07-11

**Authors:** Beatriz Almeida Sampaio, Maysa Santos Barbosa, Matheus Gonçalves de Oliveira, Manoel Neres Santos Júnior, Bruna Carolina de Brito Guimarães, Emilly Stefane Souza Andres, Ágatha Morgana Bertoti da Silva, Camila Pacheco Gomes, Rafaela de Souza Bittencourt, Thiago Macêdo Lopes Correia, Lucas Santana Coelho da Silva, Jurandir Ferreira da Cruz, Rohini Chopra-Dewasthaly, Guilherme Barreto Campos, Jorge Timenetsky, Bruno Lopes Bastos, Lucas Miranda Marques

**Affiliations:** 1Multidisciplinary Institute in Health, Federal University of Bahia, Rua Hormindo Barros 58, Vitória da Conquista 45029-094, BA, Brazil; basampaio@uesc.br (B.A.S.); matheusgonoli@gmail.com (M.G.d.O.); emilly.andres@usp.br (E.S.S.A.); agathamorgana@usp.br (Á.M.B.d.S.); rafaelasb@gmail.com (R.d.S.B.); thiagomlc@ufba.br (T.M.L.C.); guilherme.campos@ufba.br (G.B.C.); blbastos@ufba.br (B.L.B.); 2Institute of Biomedical Sciences, University of São Paulo, Avenida Professor Lineu Prestes, 2415, Butantã 05508-900, SP, Brazil; msbarbosa@uesc.br (M.S.B.); joti@usp.br (J.T.); 3Microbiology Department, State University of Santa Cruz, Rod. Jorge Amado, Km a6, Ilhéus 45662-900, BA, Brazil; neres.manoel@hotmail.com (M.N.S.J.); brunag@ufba.br (B.C.d.B.G.); cpgomes@uesc.br (C.P.G.); lscsilva@uesc.br (L.S.C.d.S.); 4Goat Sheep Sector, Department of Plant Science and Animal Science, State University of Southwest Bahia, Estrada do Bem Querer, Km 4, Vitória da Conquista 45083-900, BA, Brazil; cruzjurandir@gmail.com; 5Department of Pathobiology, Institute of Microbiology, University of Veterinary Medicine, Veterinärplatz 1, 1210 Vienna, Austria; rohini.chopra-dewasthaly@vetmeduni.ac.at

**Keywords:** contagious agalactia, *M. agalactiae*, recombinant, subunit, vaccine

## Abstract

**Background/Objectives**: Contagious agalactia (CA) is a disease typically caused by Mycoplasma agalactiae, affecting small ruminants worldwide and being endemic in certain countries. CA causes severe economic losses due to mastitis, agalactia, and arthritis. As an alternative to existing immunoprophylactic measures, this study aimed to develop a recombinant subunit vaccine against *M. agalactiae* and evaluate its specific immune response in goats. **Methods**: Goats were divided into three groups: group 1 received recombinant proteins (P40 and MAG_1560), group 2 received formalin-inactivated *M. agalactiae*, and group 3 received Tris-buffered saline (negative control). All solutions were emulsified in Freund’s adjuvant. Animals were monitored for 181 days. IgG antibody production was assessed by ELISA, and peripheral blood mononuclear cells (PBMCs) were analyzed by real-time PCR for the expression of IL-1β, IFN-γ, IL-12, and MHC class II genes. **Results**: *M. agalactiae*-specific antibody response was observed for six months in the sera of animals from group 1. Analysis of cytokine gene expression revealed increased IL-1β mRNA levels over time in both experimental groups. In group 1, IFN-γ mRNA levels increased with P40 stimulation and decreased with MAG_1560. IL-12 mRNA expression decreased over time in group 1 with P40 stimulation, whereas group 2 showed increased IL-12 expression for both proteins. MHC-II expression was stimulated in both groups. **Conclusions**: The recombinant proteins induced antibody production and cytokine expression, demonstrating immunogenic potential and supporting their promise as vaccine candidates capable of eliciting both humoral and cellular immune responses against *M. agalactiae*.

## 1. Introduction

Contagious agalactia (CA) is a disease that affects goats and sheep, first reported in Italy in 1816. However, its contagious nature was not recognized until years later, when it was observed that animals grazing in the exact location as an infected herd were also affected [1,2]. Although *Mycoplasma agalactiae* (*Ma*) is the classic etiological agent, the disease can be caused by three other mycoplasmal species, namely *Mycoplasma capricolum* subsp. *capricolum* (*Mcc*), *Mycoplasma mycoides* subsp. mycoides large colony type (*Mmm*LC), and *Mycoplasma putrefaciens* (*Mp*) [3].

The most common routes of mycoplasma entry into the host organism are oral, respiratory, and mammary. Thereafter, it is carried through the bloodstream and established in other organs, such as mammary glands, eyes, joints, and lymph nodes, where an inflammatory response develops. With the development of bacteremia accompanied by fever, symptoms such as transient hypogalactia, abrupt and total agalactia, mastitis, arthritis, polyarthritis, and keratoconjunctivitis may present [4,5]. Although antibiotics have been used as therapeutic agents, it is necessary to apply an adequate therapeutic dose for a relatively long period to avoid reducing the effect of antibiotic therapy. Incorrect administration can spread disease agents in the environment and possibly select for resistant strains [3,4,5].

Current vaccination strategies against the disease are based on the application of inactivated or attenuated vaccines. In southern Europe, commercial formalin-inactivated vaccines targeting *M. agalactiae* are used more frequently, whereas in Turkey, the application of attenuated vaccines is more prevalent [6]. The application of inactivated vaccines must be repeated after short periods since antibody stimulation has lower titers and is less persistent [7]. With attenuated vaccines, the articular lesions disappear, but the animals continue to excrete the pathogen in their milk for several months, even when it appears normal. Healthy animals, when vaccinated as a preventive measure, do not show any generalized infection or clinical signs, although temporary infection in the udder can result in some cases [5]. In Brazil, although contagious agalactia caused by *M. agalactiae* has been registered [8,9], vaccines are not available on the market, and there is no legislation regarding their use [10].

Although the mechanisms of persistence and infection of *M. agalactiae* are unknown, there is limited knowledge related to its surface antigenic diversity that contributes to its pathogenicity, including immunodominant surface lipoproteins with high-frequency switching, which are the Vpmas (variable proteins of *M. agalactiae*) [11], P40 [12], P48 [13], MAG_1560, and MAG_6130 [14]. Identifying and characterizing immunogenic proteins will enable the development of new diagnostic and prophylactic tools against contagious agalactia. For several other animal mycoplasmoses, studies have presented recombinant proteins with antigenic and immunogenic profiles for the possible composition of subunit vaccines, such as *M. hyopneumoniae* [15], *M. mycoides* subsp. *mycoides* [16], and *Mycoplasma ovipneumoniae* [17].

Two novel antigenic membrane proteins (MAG_1560 and MAG_6130) of *M. agalactiae*, identified and characterized by our research group, as well as P40 [12], have been previously evaluated for their antigenicity. They were identified in the sera of goats and sheep naturally infected with *M. agalactiae* and had exhibited potential immunogenicity, as evidenced by the production of IgG antibodies in rabbits [14]. In this study, we focused on analyzing the immune response of MAG_1560 and P40 proteins in goats to develop them as potential antigens for formulating a recombinant subunit vaccine against CA.

## 2. Materials and Methods

### 2.1. Expression and Purification of P40 and MAG_1560 Recombinant Proteins

In this study, two *M. agalactiae* antigenic recombinant proteins were used: P40 (WP_011949418.1; 42 kDa) [12] and MAG_1560 (WP_011949336.1; 32 kDa) [14]. For protein expression, the transformed *E. coli* BL21 Star™ (DE3) One Shot^®^ (Invitrogen^TM^) strain containing the vector pET-28a (+) with the gene of interest was initially cultured in a pre-inoculum (1:100) in LB medium (Luria-Bertani) containing kanamycin and incubated overnight with shaking at 37 °C, as described previously [14]. Subsequently, the pre-inoculum was brought to a volume of 1 L and incubated for 3 h until an optical density (OD600) of between 1.0 and 1.5 was reached. Protein expression was induced with 1 mM IPTG (isopropyl β-D-1-thiogalactopyranoside; Invitrogen^TM^), and the inoculum was incubated under shaking at 17 °C overnight. Cells were obtained from centrifugation, and the pellet was homogenized in Tris-buffered saline (TBS; 0.1 M Tris, 0.5 M NaCl, 10% glycerol, pH 8.5) with protease inhibitors and lysed by sonication. The sample was centrifuged after lysis, and the supernatant containing the proteins of interest was collected and filtered through a 0.22 µm membrane. The filtered solution was purified on a column containing a nickel-chelating resin (HisTrap HP, GE Healthcare Bio-Sciences Corp, San Ramon, CA, USA) with varying imidazole concentrations (20 mM–1 M). Proteins were visualized by 12% SDS-PAGE, confirmed by western blotting using the primary antibody against the 6x-His Epitope Tag (Invitrogen™), and dialyzed (SnakeSkin^TM^ Dialysis Tubing, 10 K MWCO, 22 mm, Thermo Scientific, Waltham, MA, USA) to remove residual imidazole. The proteins were concentrated using an Amicon^®^ concentrator tube. Protein quantification was performed using the Bradford method with a plate reader (iMark™ Microplate Absorbance Reader) at a 595 nm wavelength.

### 2.2. Mycoplasma Agalactiae Inactivation

The *M. agalactiae* GM 139 strain was cultivated for 48 h at 37 °C in a supplemented SP4 medium. The culture was centrifuged at 10,000× *g* for 30 min, and the pellet was resuspended in PBS. Serial dilution of the culture with plating was performed to determine CFU/mL. Before inactivation of the microorganisms, the protein concentration was determined using the Pierce BCA Protein Assay kit (Thermo Scientific, São Paulo, Brazil) according to the manufacturer’s instructions. The sample, resuspended in PBS, was inactivated by incubation at 37 °C for 16 h with 0.5% formaldehyde [18]. After incubation, the cells were washed, centrifuged under the same conditions (10,000× *g* for 30 min), and resuspended in Tris-buffered saline.

### 2.3. Membrane Protein Extraction

Based on the methodology of Rawadi and Roman-Roman [19] with alterations, membrane proteins from *M. agalactiae* were extracted by phase separation with the non-ionic detergent Triton X-114 (TX-114). The bacteria were cultured as described above and then centrifuged. The sediment obtained after removing the residual culture medium was used for extraction. The cell solution was resuspended in Tris-EDTA (50 mM Tris, pH 8.0; 0.15 M NaCl; 1 mM EDTA) containing 2% TX-114 and incubated for 3 h at 4 °C. Subsequently, the solution was incubated for 30 min at 37 °C to facilitate phase separation, centrifuged, and then incubated on ice for 5 min. The last process was repeated three times with the addition of Tris-EDTA buffer without detergent to remove proteins from the aqueous phase of the detergent. Tris-EDTA buffer was added to the final TX phase, and 2.5 times the volume of absolute ethanol was added for precipitation of membrane proteins, followed by overnight incubation at −20 °C. After incubation, the solution was centrifuged, and the precipitate was recovered and resuspended in PBS. The membrane protein concentration was quantified using the Bradford method.

### 2.4. Animal Immunization and Experimental Groups

Two different immunization solutions were prepared for immunizing the animals. Vaccine A, the protein immunizer, contained 100 µg of total recombinant protein (50 µg MAG_1560 + 50 µg P40) in 1 mL of Tris-buffered saline. Vaccine B contained 1 mL of formalin-inactivated *M. agalactiae* at 109 CFU/mL. In the first immunization, vaccines A and B were homogenized with the addition of 1 mL of Freund’s complete adjuvant (FCA) (1:1); in the second dose, the exact formulation was used but with Freund’s incomplete adjuvant (FIA) (1:1). The final volume of the inoculation solution was 2 mL. Immunization was performed in female Boer goats (*n* = 12) at five months of age. The use of animals was approved by the Animal Use Ethics Committees (CEUA) of the Federal University of Bahia, Multidisciplinary Institute of Health, Campus Anísio Teixeira (protocol 077/2019). During the experiment, the animals were kept on the campus of the State University of Southwest Bahia (UESB), fitted with ear tags for individual identification, and fed cut grass, hay, and water ad libitum. Goats were examined and evaluated for the presence of *M. agalactiae* by PCR [20]. Those considered negative were used in the experiment. The goats were randomly divided into three groups (*n* = 4): (I) inoculated with vaccine A (group 1), (II) inoculated with vaccine B (group 2), and (III) a negative control group (group 3) inoculated with Tris-buffered saline and adjuvant (FCA or FIA). The sample size of animals per group was selected based on previous studies evaluating immune responses in ruminants [21,22]. Immunization was performed via the subcutaneous route with a dose booster (Tris-saline, recombinant protein, or inactivated *M. agalactiae*) after 28 days.

The animals were evaluated for six months by collecting blood in sterile tubes with and without EDTA from each goat by venipuncture of the external jugular vein. The first collection was performed at time zero (before immunization) and at 7, 14, 21, 42, 56, 70, 91, 104, 128, 168, and 181 days after immunization. The animals were monitored daily, and the procedures were carried out by trained staff under the supervision of a veterinarian. To minimize stress, the animals were handled carefully, and blood was collected using manual restraint.

### 2.5. Determination of Specific IgG Production Levels by Indirect ELISA

The production of antibodies in response to proteins or inactivated *M. agalactiae* was determined using an indirect ELISA assay. This technique was previously standardized concerning the amount of antigen used. Microplate wells were coated with total protein extract of *M. agalactiae* (1 µg/well in carbonate-bicarbonate buffer, pH 9.6) obtained by lysis with RIPA buffer, P40 recombinant protein, MAG_1560 (50 ng/well in carbonate-bicarbonate buffer, pH 9.6), or *M. agalactiae* membrane proteins (1 µg/well in carbonate-bicarbonate buffer, pH 9.6), and incubated at 4 °C for 16 h. Reactions were blocked with PBS containing 0.05% Tween-20 (PBS-T) and 10% skim milk after the wells were washed with PBS-T. Sera collected from animals in each group were diluted (1:200) in PBS-T containing 5% skim milk and incubated at 37 °C. As a negative control, serum from a group 3 animal was used. Afterwards, the wells were washed with PBS-T, and the secondary antibody anti-goat IgG (Rabbit anti-goat IgG, HRP conjugate – Invitrogen, Thermo Fisher, São Paulo, Brazil) diluted (1:2000) in PBS-T containing 5% skim milk was added. The plates were incubated at 37 °C, and the reaction was developed using o-phenylenediamine dihydrochloride (OPD, Thermo Scientific, São Paulo, Brazil) and hydrogen peroxide. The reaction was stopped using sulfuric acid (H_2_SO_4_). The plate reading was performed at 492 nm using an ELISA reader (iMark™ Microplate Absorbance Reader).

### 2.6. Avidity Test for IgG Antibodies

The avidity test for specific antibodies was performed following the methodology described by Barbosa et al. [14]. For this assay, serum from animals (groups 1 and 2) collected 181 days after the first immunization was used. In brief, the serum dilution yielding an OD at 450 nm of 1.0 was first determined via indirect ELISA. The procedure for the affinity test was similar to the indirect ELISA described above, except that, after incubation with the serum, the plate was incubated with varying concentrations of ammonium thiocyanate (up to 8 M) or PBS alone for 15 min. The plate was subsequently incubated with anti-goat IgG (1:2000) and visualized using TMB (tetramethylbenzidine). The plate reading was performed at 450 nm using an ELISA reader (Kasuaki^TM^).

### 2.7. Colony Immunoblotting Analysis

Serum collected from the immunized animals was analyzed to detect *M. agalactiae* colonies. The *M. agalactiae* GM139 strain was cultured on plates of SP4 medium supplemented with agar, and the resulting colonies were transferred to nitrocellulose membrane discs (Bio-Rad) placed on the agar surface. The membranes were washed with PBS-Tween and incubated overnight at 4 °C with serum (1:200) from group 1 and 2 animals, collected at pre-immunization and 56- and 181-day post-immunization. After incubation, the membranes were washed three times and then incubated for 1 h and 30 min with a peroxidase-conjugated secondary antibody specific for goat IgG (Rabbit anti-goat IgG, HRP conjugate– Invitrogen, Thermo Fisher, São Paulo, Brazil), diluted 1:2000. The membranes were subsequently washed and visualized using DAB and hydrogen peroxide.

### 2.8. Analysis of Gene Expression of Inflammatory Markers Triggered by In Vitro Stimulation of Peripheral Blood Mononuclear Cells (PBMCs)

A comparative analysis of inflammatory marker responses (IL-1β, IFN-γ, IL-12, and major histocompatibility complex class II—MHC II) was performed in peripheral blood cells obtained from animals before immunization (time zero) and at 56 and 168 days post-immunization. PBMCs from the test groups were isolated for in vitro stimulation of goat cells with a 2-h incubation period [23]. Five milliliters of peripheral blood were collected from each animal. Isolation was performed using Histopaque-1077^®^ (density: 1.077 g/mL; Sigma-Aldrich, São Paulo, Brazil). The mononuclear cells of each animal were collected, and a pool of PBMCs was formed from the experimental group. After washing, RPMI-1640 medium (Gibco, Themo Fisher, São Paulo, Brazil) supplemented with 10% fetal bovine serum and 100 U/mL penicillin was added to the cell pellet for resuspension. The process proceeded with counting cells in a Neubauer chamber using Trypan Blue. The cell volume was adjusted to a concentration of 2 × 10^5^ cells/mL, and the cell pool was added in triplicate to a 24-well plate. Cells were stimulated with inactivated *M. agalactiae* at 10^5^ CFU/mL and with P40 and MAG_1560 proteins at 1 and 2 µg/mL concentrations. Tris-buffered saline was added as a negative control, and LPS was used as a positive control. Cells were incubated at 37 °C with 5% CO2 for 2 h. A cell suspension was obtained by centrifugation of the supernatant, and cells adhering to the plate were removed using trypsin. The cells were stored in RNA*later* (Invitrogen, Thermo Fisher, São Paulo, Brazil).

### 2.9. RNA Extraction, cDNA Synthesis, and Gene Expression

RNA was extracted using TRIzol™ (Invitrogen, Brazil) according to the manufacturer’s instructions and purified using the PureLink RNA Mini Kit (Invitrogen, Thermo Fisher, São Paulo, Brazil). cDNA was synthesized from the extracted mRNA using the SuperScript IV Reverse Transcriptase kit (Invitrogen, Thermo Fisher, Brazil). The gene expression of the major histocompatibility complex class II (MHC-II) [24] and of the cytokines IL-1β [25], IFN-γ [26], and IL-12 [27] was analyzed. β-actin [28] was used as an endogenous control. RT-qPCR reactions were performed on the StepOnePlus PCR Real-Time System platform (Applied Biosystems™, Thermo Fisher, São Paulo, Brazil) with Power SYBR^®^ Green Master Mix (Applied Biosystems™, Thermo Fisher, São Paulo, Brazil). Data were analyzed using the comparative CT method (2^−ΔΔCt^).

### 2.10. Statistical Analysis

Statistical analyses were performed using GraphPad Prism version 5.0 (GraphPad Software version 10.0, USA) and the nonparametric Kruskal-Wallis test with Dunn’s post-hoc test to differentiate the ELISA groups. The nonparametric Mann-Whitney U test was used for gene expression analysis to compare the two groups. Statistically significant differences were considered at *p* ≤ 0.05, with a 95% confidence interval. Values are presented as mean ± standard error of the mean.

## 3. Results

### 3.1. Purified P40 and MAG_1560

The P40 and MAG_1560 proteins were successfully expressed in a heterologous *E. coli* system, purified using nickel resin affinity chromatography, and visualized by Western blotting (Figure 1). The uncropped western blots are presented in Appendix A.

### 3.2. The Recombinant Vaccine Induced a High Rate of Specific IgG Antibody Production

The production of specific IgG antibodies by each group was evaluated over 181 days in response to the recombinant proteins introduced from immunization (P40 and MAG_1560), total protein extract, and membrane proteins of *M. agalactiae*. In group 1, the animals’ immune systems generated a high level of antibodies to both inoculated proteins (Figure 2A,B), which persisted throughout the analysis period, demonstrating their immunogenic capacity. The group immunized with inactivated *M. agalactiae* (group 2) produced specific antibodies against the MAG_1560 protein (Figure 2B), showing a statistically significant difference (*p* < 0.05) compared to group 3, which was immunized with only buffer and adjuvant. Despite the low values presented in group 1 in response to the total protein extract (Figure 2C), a statistically significant difference (*p* < 0.05) was observed between groups 1 and 3 at 42, 128, 168, and 181 days after immunization. Serum from both groups 1 and 2 showed antibody production during 181 days in response to *M. agalactiae* membrane protein (Figure 2D).

The effect of ammonium thiocyanate on the avidity of antibody-antigen binding is illustrated in the graphs in Figure 3A,B. In group 1, for both proteins, 50% inhibition of binding occurred at concentrations above 1 M ammonium thiocyanate. The antibodies present in the serum of group 1 animals exhibited greater avidity, as higher concentrations of ammonium thiocyanate were required to disrupt the interaction compared to group 2 (Figure 3C).

### 3.3. Colony Recognition by Specific Antibodies Over Time Post-Immunization

Colony immunoblotting (Figure 4) revealed that the specific antibodies present in the serum of group 1 animals (immunized with recombinant proteins) were capable of recognizing *M. agalactiae* colonies at 56 and 181 days after the first dose, with a greater staining intensity compared to pre-immune serum. In group 2 (immunized with inactivated *M. agalactiae*), there was recognition at 56 days after immunization compared to the pre-immune serum. At 181 days post-immunization, group 2 exhibited an extremely reduced staining intensity compared to group 1. Overall, the serum from group 1 demonstrated a higher staining intensity, suggesting an enhanced ability of these antibodies to detect colonies six months post-immunization. All original blot membranes are presented in Appendix A.

### 3.4. The P40 and MAG_1560 Proteins Can Alter the Gene Expression of Cytokines and MHC-II in Different Groups

The gene expression in PBMCs isolated from the animals was evaluated before inoculation (0 day), 56 days, and 168 days after immunization with *in vitro* stimulation in cell culture, with 2 h of incubation with P40 and MAG_1560. Cells from the group immunized with vaccines A and B in response to P40 stimulation at different concentrations showed a statistically significant increase in the expression of the pro-inflammatory cytokine IL-1β (Figure 5A,C) at 56 and 168 days post-immunization compared to before immunization. The increase in expression also occurred in group 2 in response to P40 at 56 and 168 days, compared to the negative control, except at the 2 µg/mL concentration, which did not show a statistically significant difference at 56 days. In response to MAG_1560, group 1 showed a substantial increase in the concentration of 1 µg/mL at the time of immunization for the IL-1β gene (Figure 5B). Group 2 showed increased IL-1β gene expression at both MAG_1560 concentrations (Figure 5D).

The P40 protein also induced an increase in gene expression of IFN-γ in group 1 cells at a higher concentration (2 µg/mL) of the protein, as stimulated at 168 days (Figure 6A). In response to MAG_1560 (Figure 6B), at 56 days, group 1 showed a reduction in gene expression with a statistically significant difference at a protein concentration of 1 µg/mL. At 168 days post-immunization, there was no induction or reduction in cytokine gene expression for either concentration of MAG_1560 protein. Group 2 demonstrated a decrease in gene expression over time, with stimulation of both proteins at concentrations different from those before immunization (Figure 6C,D), except for 1 µg/mL of MAG_1560 after 168 days.

IL-12 gene expression was reduced in the post-immunization period in group 1 in response to P40 (Figure 7A). However, 168 days later, compared to the negative control, the cells showed an induction of cytokine expression. Group 1 showed a statistically significant difference (*p* < 0.05) for MAG_1560 compared to pre-immunization, with induction and reduction of expression observed after stimulation with 1 and 2 µg/mL, respectively, only 56 days after immunization. P40, at its lowest concentration, induced the expression of the IL-12 gene (Figure 7C), while MAG_1560 reduced the expression (Figure 7D) in the cells of group 2, 56 days after immunization. At 168 days, group 2 demonstrated induction of IL-12 expression with both proteins (Figure 7C,D) at both concentrations, compared to the response before immunization.

The MHC-II molecule, found in antigen-presenting cells in response to P40 protein stimulation in group 1 (Figure 8A), exhibited gene expression induced at 56 days and a statistically significant reduction at 168 days. MAG_1560 (Figure 8B) induced gene expression at minimal concentrations at 56 and 168 days, but the expression was reduced at a concentration of 2 µg/mL at the last evaluation. Group 2 showed reduced expression at day 56 and induction at day 168 for both proteins at a concentration of 1 µg/mL (Figure 8C,D). In contrast, for the same group, there was a reduction in the two periods with the highest concentration of P40 (Figure 8C).

## 4. Discussion

The development of new immunoprophylactic measures as alternatives to existing ones, particularly in terms of their protective efficacy and safety, poses a challenge in controlling *M. agalactiae*. This bacterium is the primary causative agent of contagious agalactia in small ruminants, is responsible for high morbidity in animals, and is associated with economic losses in European countries and Brazil [8,29]. Current licensed vaccines (attenuated and inactivated) induce antibody titers that decline quickly [30]. The use of antigenic proteins of microorganisms, focusing on membrane proteins, is a target of the new vaccine model for the disease [14]. In this study, immunizing goats with two recombinant proteins, P40 and MAG_1560, allowed us to observe the evolution of the immune system response developed within six months post-immunization.

Membrane proteins play a crucial role in establishing natural infections caused by microorganisms, and they can induce a humoral response due to their highly immunogenic characteristics [12,31]. The persistence of specific antibodies in the immunized groups (1 and 2) was evaluated over a six-month period; the presence of antibodies against recombinant proteins, total protein extract, and membrane proteins of *M. agalactiae* was demonstrated, indicating the effectiveness of the strategy for producing high levels of antibodies. In another study, sera from animals naturally infected with *M. agalactiae* showed variable antigen recognition with antibodies against membrane proteins from 18 kDa to 80 kDa [31]. In the present study, immunization with recombinant proteins (group 1) and inactivated *M. agalactiae* (group 2) resulted in a five-fold increase in antibody production compared to the control group (group 3). In contrast, another study carried out with sheep immunized with *M. agalactiae* emulsified in oily adjuvant showed a threefold increase in production. In the previously mentioned study, the animals exhibited persistent antibody production for five months and developed immunity to challenge with the microorganism, thereby preventing clinical signs of the disease [32]. The period of antibody persistence presented in this study was similar to that in our analysis of immunization with recombinant proteins. These results indicate promising outcomes for microorganism challenges after immunization with these recombinant proteins, particularly considering the recognition of the colonies. Furthermore, these data support the probable efficacy of recombinant vaccines compared to inactivated vaccines since immunization with inactivated microorganisms in a previous study [32] induced a lower production of antibodies compared to our research. However, it was still able to prevent clinical signs of the disease.

It is essential to consider that, for the effectiveness of potent and long-lasting modulated immune responses to subunit vaccines, which are less reactive, co-administration with potent adjuvants is necessary, as the adjuvant helps increase avidity and titration [33,34]. Despite the adverse reactions often associated with its use, Freund’s adjuvant is highly effective in inducing high antibody titers and increasing cellular and humoral immune responses. Due to its reactogenicity, its application is restricted to experimental animal immunization. The complete Freund’s adjuvant (CFA) is generally used in primary immunization, while the incomplete formulation is used in booster doses to reduce adverse effects [35,36]. Antibody production was also observed following the application of experimental recombinant vaccines with oily adjuvants for other mycoplasmoses, such as *Mycoplasma mycoides* subsp. *mycoides* responsible for contagious bovine pleuropneumonia demonstrated that protection can be achieved with recombinant proteins, leading to a reduction in clinical signs of the disease [16]. Studies using recombinant proteins as a form of immunization, whether for human or veterinary applications, have reinforced the application of this vaccine model [15,16,17,37]. The data found here regarding the high production of antibodies provided by the recombinant vaccine offers an alternative to current veterinary vaccines for contagious agalactia, enabling the development of an effective and safe immune response to mycoplasmosis. This study showed a gradual increase in the expression of the IL-1β gene, with the maximum expression observed 168 days after the first immunization. In the absence of a cell wall in mycoplasmas, membrane proteins, the most abundant lipoproteins, play an essential role in immunomodulating the host’s immune system [38]. PBMCs, when stimulated in vitro with mycoplasmas, such as *M. bovis*, exhibit higher IL-1β expression compared to unstimulated cells [39]. A similar trend is observed in the transcriptome analysis of bovine mammary epithelial cell lines infected with *M. bovis*, which reveals increased expression of IL-1β and IL-6 [40]. Similarly, in our study, we observed this response with increased IL-1β mRNA expression in goat PBMCs following immunization. Members of the interleukin-1 family, such as IL-1β, are closely related to the innate immune response and act as modulators of inflammatory processes [41]. In addition, cytokine IL-1β can contribute to regulating adaptive immune responses by acting as a “licensing cytokine” in the effector functions of all memory TCD4 cell lines [42]. Upon antigen re-exposure, these cells can mount a faster and more robust response than naïve T cells, promoting the differentiation of memory B cells into plasma cells [43]. Thus, the increase in PBMC IL-1β mRNA expression over time in response to stimulation with recombinant proteins suggests that cytokine activity is present in these memory cells. The production of IL-1β can also be observed upon induction by Mycoplasma lipid-associated membrane proteins (LAMPs) derived from *M. bovis* in embryonic bovine lung cells, which activate the NF-κB pathway through TLR2 and MyD88 [23]. The same was observed for LAMPs from *M. gallisepticum* in a chick embryo fibroblast cell line [44]. Regulatory T cells (Tregs) must overcome effector cells to develop an appropriate immune response to pathogens, and the cytokines IL-1 and IL-6 mediate the resistance of these cells. The signaling of IL-1 receptors via MyD88 is an essential interaction for Th1 cells to overcome the regulatory activity of Treg cells and develop [45]. In a study evaluating the humoral and cellular immune responses of sheep immunized with attenuated *Brucella melitensis* vaccines, a predominant and lasting Treg cell response was observed in their PBMCs, suggesting an important role in reducing the protective efficacy of vaccines [46]. Regulatory T cells, as an important component of the natural response to infections, limit inflammation and may consequently reduce vaccine efficacy by attenuating host pathology associated with vaccine-induced immune reactions [47]. The increased expression of IL-1β mRNA in immunized animals in response to *M. agalactiae* antigens observed in the present study may be associated with Treg cell suppression, contributing to the development of effector cells in the immune response caused by the vaccine.

In most data, the time-dependent expression of IFN-γ mRNA in groups 1 and 2 in response to the presence of the recombinant proteins showed no difference in expression or produced lower cytokine levels compared to pre-immunization levels. In agreement with the results of our study, in vitro analysis of PBMCs from infected sheep exposed to *M. agalactiae* also showed a decrease in the expansion of IFN-γ-positive cells after 60 days of infection [48]. IFN-γ is involved in coordinating the immune response through the stimulation of macrophages, induction of antimicrobial mechanisms, antigen presentation, tissue protection, and cell differentiation. Cytokines secreted by antigen-presenting cells during the reactivation phase, such as IL-12, regulate the production of IFN-γ, and cytokines like IL-4 can downregulate [49,50]. Therefore, it is secreted mainly by activated lymphocytes, such as Th1 cells, and inhibits Th2 cell proliferation and IL-4 production [51]. Since this cytokine can be negatively regulated by the presence of IL-4, associated with a Th2 cell profile, the formation of a memory profile induced by the humoral response in the cells of group 1 is indicated here in our study. This response profile is similar to that observed in a study of mice immunized with inactivated *M. agalactiae* emulsified with FCA, which showed the production of *M. agalactiae*-specific IgG, accompanied by Th2 activation and IL-4 secretion, with relatively low co-secretion of cytokines such as IFN-γ and IL-12 [18]. It is important to consider that IFN-γ production is associated with developing type 1 helper T cells from antibody isotype switching [52]. Identifying IFN-γ levels contributes to understanding the establishment and development of response profiles over time and their role in the development of a protective immune response.

The cytokine IL-12 is primarily produced by cells such as macrophages, monocytes, neutrophils, and dendritic cells and plays a central role in inducing the Th1 response. Since microbial products strongly induce its production, among its biological functions, IL-12 is closely related to the induction of IFN-γ production [53]. This synergistic correlation was observed in group 1, 168 days post-immunization, compared to the negative control of unstimulated cells, which showed a significant increase in IL-12 and IFN-γ mRNA expression. Similarly, both cytokines were increased in spleen mononuclear cells from mice immunized with a DNA vaccine of the P48 protein of *M. agalactiae* [54]. In response to the stimulus of both proteins, an increase in IL-12 mRNA expression was observed over time. In response to re-exposure to recombinant antigenic proteins in 2 h of culture, the immunized goat PBMCs in our study produced a faster response with higher levels of IL-12 expression. The response of memory T cells, which are capable of modulating the memory immune response when exposed to antigens, differs from that of naïve T cells in that they can immediately produce cytokines associated with the ability to recall and fight pathogens quickly [54]. PBMCs can also be stimulated to produce IFN-γ in response to IL-12 produced by B cells, thereby simultaneously enhancing the T-cell response [55,56]. At the beginning of the adaptive immune response with specificity for pathogens, MHC molecules play a central role in presenting peptide antigens that are processed and presented to T cells [57]. MHC-II molecules play a crucial role in antigen presentation to CD4+ T lymphocytes, and their expression is associated with antigen-presenting cells (APCs), including macrophages, monocytes, dendritic cells, and B lymphocytes [58]. Mononuclear cells from groups 1 and 2 at 56 and 168 days post-immunization showed an increase in MHC-II mRNA expression compared to cells without previous contact with the antigen (recombinant proteins). Through MHC antigen recognition, memory T cells can develop a potent and enhanced recall response, a critical factor for long-term immunity [59]. B cells can also express MHC-II molecules on their surface, thereby establishing contact with TCD4 cells through the peptide-MHC-II complex, which contributes to their proliferation and differentiation [59]. The increased expression of this molecule enables the immune system to recognize proteins, either by T or B cells, which are responsible for orchestrating the response in sequence. In a study evaluating the immunohistochemistry of lung lesions in goats infected with *M. agalactiae* and *M. bovis*, an increase in mononuclear cells expressing MHC-II was observed, suggesting the involvement of signaling in antigen presentation to cells through cellular or humoral responses [60].

## 5. Conclusions

Briefly, the present study demonstrated that the immunized groups, with recombinant protein and inactivated *M. agalactiae*, were able to generate a high production of antibodies for at least six months, in addition to presenting serological responsiveness to the total extract and membrane protein portion. We also observed that PBMC responsiveness to the stimulus generated by the proteins increased over time, accompanied by an increase in the mRNA expression of the proinflammatory cytokine IL-1β. The antigenic presentation was confirmed by the expression of MHC-II and other cytokines, including IFN-γ and IL-12, which indicates the ambiguity of the induced response profile (both cellular and humoral). Although the results demonstrated the immunogenicity of the recombinant formulation, the study was conducted in an experimental model without pathogen challenge and involved only a single adjuvant in the vaccine formulation. Further studies should aim to elucidate the complementarity of immune responses by evaluating additional response mechanisms, testing different adjuvants in the formulation, and assessing vaccine efficacy. This is the first report evaluating recombinant proteins from *M. agalactiae* in goats, and the data presented here indicate the promise of recombinant proteins in composing a recombinant vaccine as an alternative to existing vaccines for contagious agalactia. This study also contributes to the understanding of immunomodulation that provides immunity through the membrane proteins of *M. agalactiae*.

## Figures and Tables

**Figure 1 vaccines-13-00746-f001:**
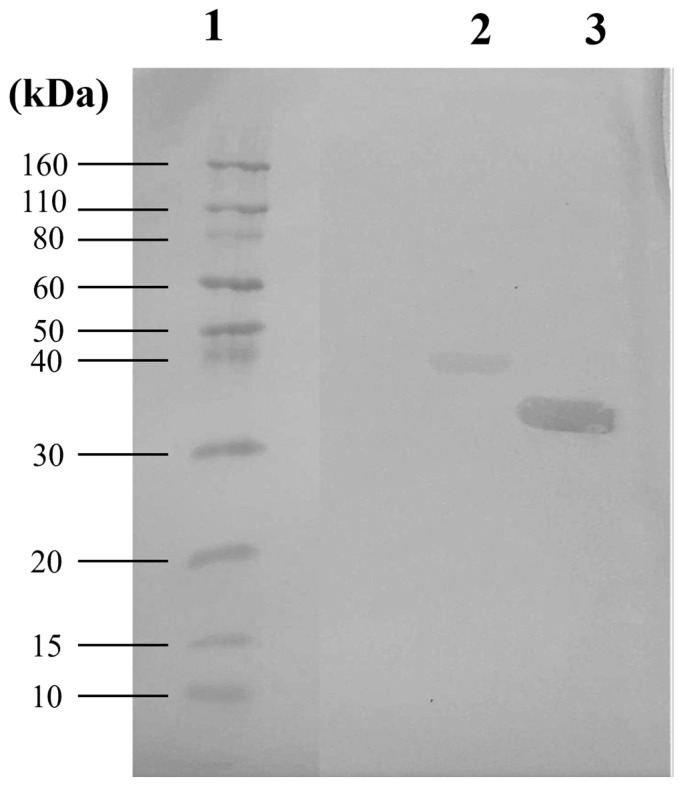
Purified recombinant proteins visualized by western blotting. Column 1: Ladder: Novex^®^ Sharp Pre-stained Protein Standard; Column 2: P40 (42 kDa); Column 3: MAG_1560 (32 kDa).

**Figure 2 vaccines-13-00746-f002:**
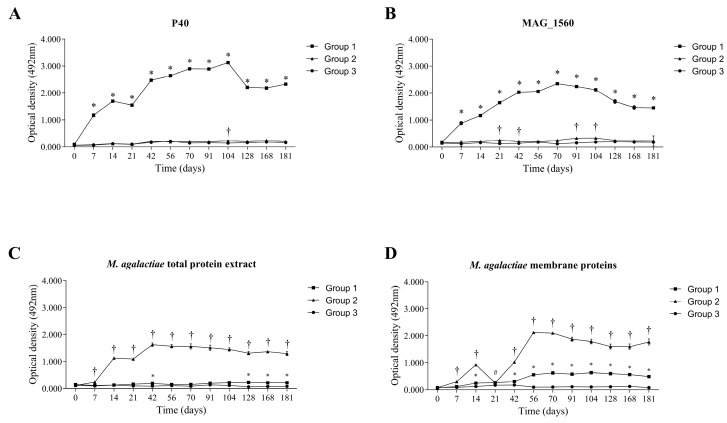
Mean values (optical density) for serum IgG antibodies determined by indirect ELISA in vaccinated animals (groups 1 and 2) and negative control (group 3). (**A**) Production of antibodies in response to P40, (**B**) MAG_1560, (**C**) *M. agalactiae* total protein extract, and (**D**) *M. agalactiae* membrane proteins. The groups were compared using the non-parametric Kruskal-Wallis test and Dunn’s post-hoc test. Statistical significance (*p* < 0.05) is represented by symbols when (#) is different from group 3, (*) is different from groups 2 and 3, and (†) is different from groups 1 and 3.

**Figure 3 vaccines-13-00746-f003:**
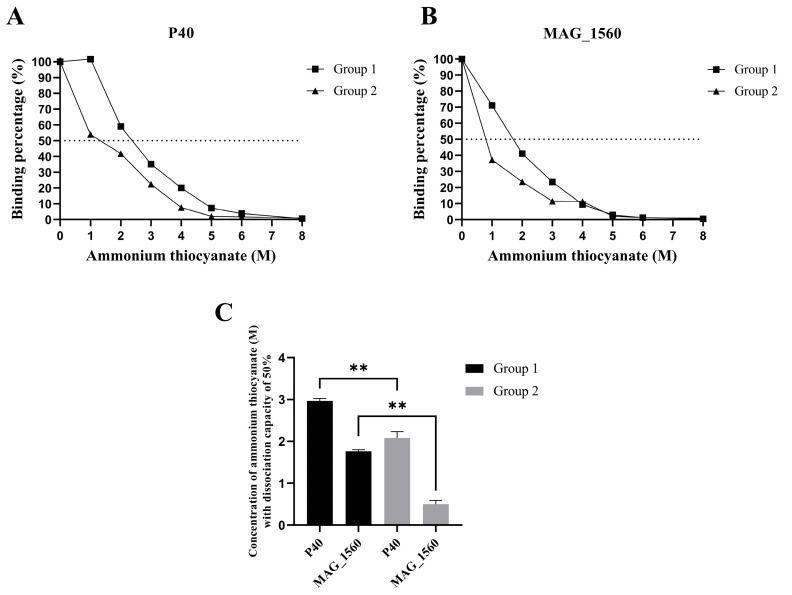
Avidity profiles of antibodies produced post-immunization. Percentage of antibodies binding to (**A**) P40 and (**B**) MAG_1560 in increasing molarities of ammonium thiocyanate. (**C**) The molarity of ammonium thiocyanate is capable of dissociating 50% of the antibody-antigen binding. Values represent means ± SD. ** *p* < 0.05. (one-way ANOVA with Bonferroni post-test).

**Figure 4 vaccines-13-00746-f004:**
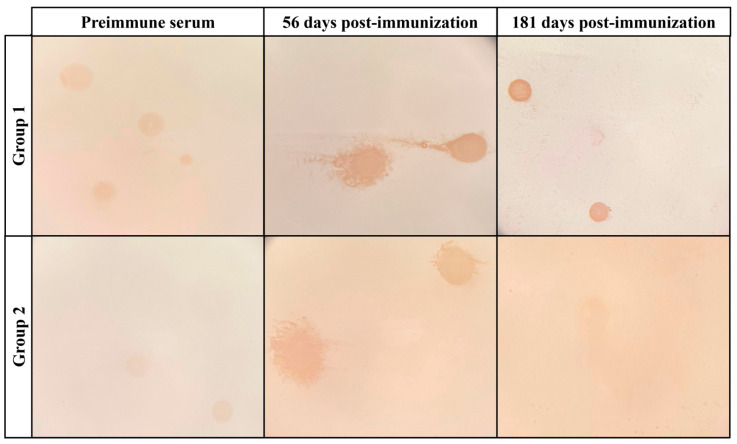
Antibody detection of *M. agalactiae* colonies over time. Detection of *M. agalactiae* strain GM139 colonies by specific antibodies present in the serum of group 1 and group 2 animals at pre-immunization and 56 and 181 days post-immunization. Recognition is indicated by increased staining intensity, represented by the brown coloration. Images were captured using a stereoscope.

**Figure 5 vaccines-13-00746-f005:**
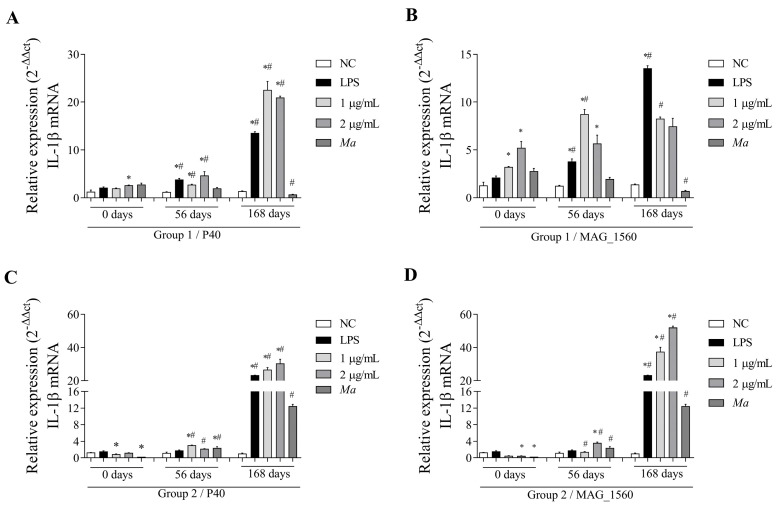
IL-1β gene expression in PBMCs isolated from vaccinated goats (groups 1 and 2) at 0 (pre-immunization), 56, and 168 days post-immunization. Cells were stimulated with 1 and 2 µg/mL of P40 and MAG_1560 recombinant proteins for 2 h. (**A**) IL-1β gene expression in P40-stimulated group 1 cells. (**B**) IL-1β gene expression in group 1 cells stimulated with MAG_1560. (**C**) IL-1β gene expression in P40-stimulated group 2 cells. (**D**) IL-1β gene expression in group 2 cells stimulated with MAG_1560. When comparing stimuli between times, the nonparametric Mann-Whitney test was used. Statistical differences were considered when *p* < 0.05 and are represented by (#) when different from the same group in the time before immunization (0 days). The nonparametric Kruskal-Wallis test, followed by Dunn’s post hoc test, was used for comparison with the negative control at different time points (0, 56, and 168 days). Statistical differences were considered when *p* < 0.05 and are represented by (*) when different from the respective negative control.

**Figure 6 vaccines-13-00746-f006:**
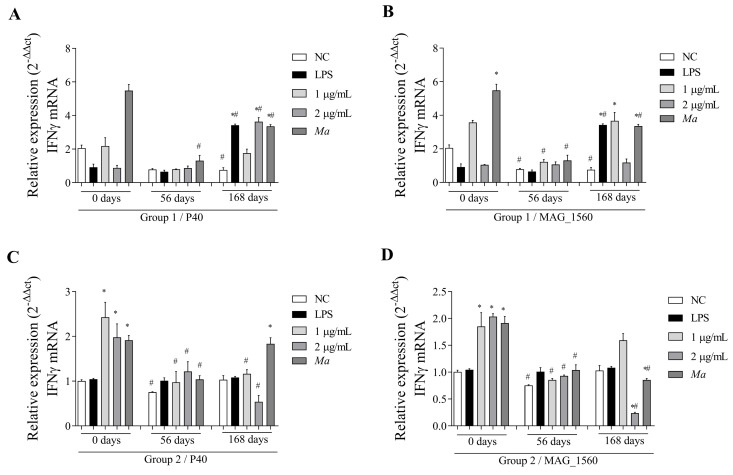
IFN-γ gene expression in PBMCs isolated from vaccinated goats (groups 1 and 2) at 0 (pre-immunization), 56, and 168 days post-immunization. Cells were stimulated with 1 and 2 µg/mL of P40 and MAG_1560 recombinant proteins for 2 h. (**A**) IFN-γ gene expression in P40-stimulated group 1 cells. (**B**) IFN-γ gene expression in group 1 cells stimulated with MAG_1560. (**C**) IFN-γ gene expression in P40-stimulated group 2 cells. (**D**) IFN-γ gene expression in group 2 cells stimulated with MAG_1560. When comparing stimuli between times, the nonparametric Mann-Whitney test was used. Statistical differences were considered when *p* < 0.05 and are represented by (#) when different from the same group in the time before immunization (0 days). The nonparametric Kruskal-Wallis test, followed by Dunn’s post hoc test, was used for comparison with the negative control at different time points (0, 56, and 168 days). Statistical differences were considered when *p* < 0.05 and are represented by (*) when different from the respective negative control.

**Figure 7 vaccines-13-00746-f007:**
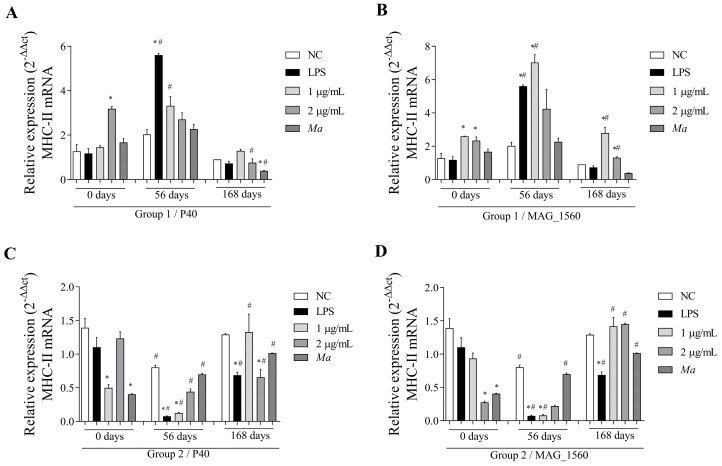
IL-12 gene expression in PBMCs isolated from vaccinated goats (groups 1 and 2) at 0 (pre-immunization), 56, and 168 days post-immunization. Cells were stimulated with 1 and 2 µg/mL of P40 and MAG_1560 recombinant proteins for 2 h. (**A**) IL-12 gene expression in P40-stimulated group 1 cells. (**B**) IL-12 gene expression in group 1 cells stimulated with MAG_1560. (**C**) IL-12 gene expression in P40-stimulated group 2 cells. (**D**) IL-12 gene expression in group 2 cells stimulated with MAG_1560. When comparing stimuli between times, the nonparametric Mann-Whitney test was used. Statistical differences were considered when *p* < 0.05 and are represented by (#) when different from the same group in the time before immunization (0 days). The nonparametric Kruskal-Wallis test, followed by Dunn’s post-hoc test, was used for comparison with the negative control at different time points (0, 56, or 168 days). Statistical differences were considered when *p* < 0.05 and are represented by (*) when different from the respective negative control.

**Figure 8 vaccines-13-00746-f008:**
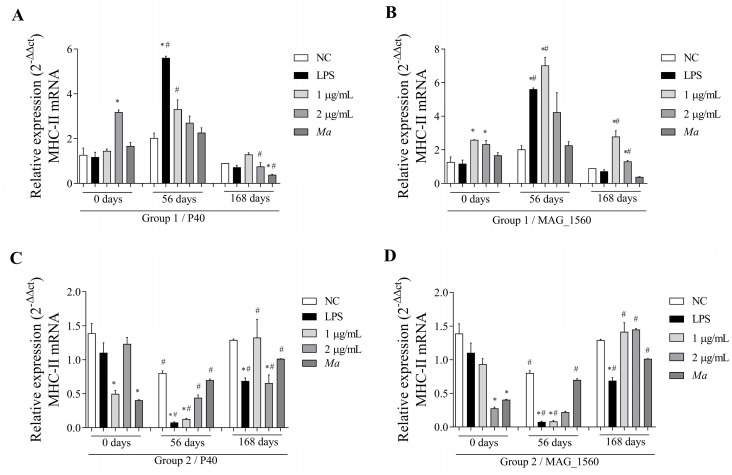
MHC-II gene expression in PBMCs isolated from vaccinated goats (groups 1 and 2) at 0 (pre-immunization), 56, and 168 days post-immunization. Cells were stimulated with 1 and 2 µg/mL of P40 and MAG_1560 recombinant proteins for 2 h. (**A**) MHC-II gene expression in P40-stimulated group 1 cells. (**B**) MHC-II gene expression in group 1 cells stimulated with MAG_1560. (**C**) MHC-II gene expression in P40-stimulated group 2 cells. (**D**) MHC-II gene expression in group 2 cells stimulated with MAG_1560. When comparing stimuli between times, the nonparametric Mann-Whitney test was used. Statistical differences were considered when *p* < 0.05 and are represented by (#) when different from the same group in the time before immunization (0 days). The nonparametric Kruskal-Wallis test, followed by Dunn’s post-hoc test, was used for comparison with the negative control at different time points (0.56, or 168 days). Statistical differences were considered when *p* < 0.05 and represented by (*) when different from the respective negative control.

## Data Availability

The datasets used and/or analyzed during the current study are available from the corresponding author on reasonable request.

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
