# Peer review of "Immune Responses Induced by Recombinant Membrane Proteins of Mycoplasma agalactiae in Goats"

_vaccines, 2025, doi:10.3390/vaccines13070746_

Round 1

Reviewer 1 Report

Comments and Suggestions for Authors
  1. The grouping format in the image should be the same.

  1. In the detection of mRNA expression of related immune factors, was the corresponding protein expression detected.

  1. The reference literature is recommended to update the research results within the last three years, and many of the manuscripts are relatively old.

  1. The manuscript should supplement the limitations of the article and the direction of future research.

Author Response

Comment 1:
The grouping format in the image should be the same.

Response 1: Thank you for pointing this out. We agree, and changes were made to the grouping format in Figure 3 (page 8) to match Figure 2 (page 8).

Comment 2: In the detection mRNA expression of related immune factors, was the corresponding protein expression detected.

Response 2: Thank you for pointing that out. Unfortunately, due to resource limitations we were not able to perform additional tests, such as cytokine ELISA, to confirm the corresponding expression. However, although we are aware that confirming the corresponding protein would be ideal, the gene expression analysis, which is a more accessible approach in this context, provided positive initial indications for future experiments aimed at evaluating the immune response that we intend to conduct.

Comment 3: The reference literature is recommended to update the research results within the last three years, and many of the manuscripts are relatively old.

Response 3: We appreciate the reviewer’s insightful observation. We agree and understand the importance of updating references, therefore, new information (lines: 81, 425, 444, 448, 452, 466, 478 and 508) supported by updated citations has been added (references: 2, 17, 35, 36, 40, 41, 43, 47, 50, 55 and 56). However, it is important to highlight the limitations within the field of mycoplasmology and studies involving goats. Unfortunately, there is a lack of regularly updated publications closely related to this topic. For this reason, in order to provide a more comprehensive presentation, discussion, and comparison of our data, we had to include some references that are older than three years.

Comment 4: The manuscript should supplement the limitations of the article and the direction of future research

Response 4: We thank the reviewer for the valuable comment. This point is indeed important. Our limitations and direction of future research was added in lines 536-541. Thus, we emphasize that it was not possible to perform the animal challenge or to evaluate more than one adjuvant in the formulation. Furthermore, complementary studies are necessary to elucidate the mechanisms of the immune response—such as the analysis of the final expressed protein, rather than gene expression alone—as well as to test different adjuvants and assess vaccine efficacy.

4. Response to Comments on the Quality of English Language

Not applicable.

5. Additional clarifications

Based on the positive results obtained in this study, new immunization trials using the recombinant proteins are already being conducted to better understand the immune response and assess the vaccine's efficacy as described in future directions.

Reviewer 2 Report

Comments and Suggestions for Authors

The study is devoted to a comparative analysis of the complex effect of recombinant Mycoplasma agalactiae surface proteins and inactivated cell lysates on the immune system of goats. The experiments conducted convincingly demonstrate the advantage of protein vaccines over inactivated cells (whole cell lysates). They cause a stronger and more stable immune system response. Their production may slightly increase the cost of vaccine production, but the economic effect of more active counteraction to contagious agalactia may more than cover any costs of insurance against its occurrence. The work is interesting, important not only for the economy and agriculture of Brazil, but also for all regions endemic for this disease. According to the authors, the article is the first report of its kind, and a number of experiments will be required to advance in the creation of a new type of recombinant vaccine. The manuscript may be published after revision.

Major points

- I don't like the image of the Western blot analysis (Figure 1). Are there any other images? Perhaps the authors have working versions of repeated experiments? Or is there a way to redo it? So far, it seems (possibly falsely) that the spots look unnatural. Please forgive me for being picky and convince the reviewer otherwise. We are all working together to improve the quality of the content.

- The authors have chosen a good and visual method with mycoplasma colonies (Figure 4). However, for the sake of completeness, it would be good not only to select a few bright and spectacular colonies, but to see a broader picture. Are there photographs of whole membranes, where a larger number of colonies are presented? It would be good to present them to the public at least in additional materials (a file with the originals of blots, gels, membranes).

- How do the authors explain the difference in IFN-γ gene expression when stimulated by recombinant proteins? There, depending on the protein (and its concentration), the opposite effect is obtained on day 168 (Figure 6 A, B, 1 µg or 2 µg of the protein).

Minor points

- What strain of M. agalactiae was used in the work? This should be stated in the materials and methods, not just in the legend to Figure 4.

- In Figure 2, at a quick glance, the identical letter designations of the groups of vaccinated animals and parts of the drawing corresponding to different vaccination options are confusing. Perhaps some things should be renamed; for example, renaming the groups 1, 2, 3 in the figure and in the corresponding places in the manuscript? At the discretion of the authors.

Author Response

Comment 1:
I don't like the image of the Western blot analysis (Figure 1). Are there any other images? Perhaps the authors have working versions of repeated experiments? Or is there a way to redo it? So far, it seems (possibly falsely) that the spots look unnatural. Please forgive me for being picky and convince the reviewer otherwise. We are all working together to improve the quality of the content.

Response 1: Thank you for pointing this out. A clearer image of the Western blot has been added to the main text, and a full membrane image has been included in the supplementary material. We are including here the corresponding color version of the image to improve visualization and address any potential questions.

Comment 2: The authors have chosen a good and visual method with mycoplasma colonies (Figure 4). However, for the sake of completeness, it would be good not only to select a few bright and spectacular colonies, but to see a broader picture. Are there photographs of whole membranes, where a larger number of colonies are presented? It would be good to present them to the public at least in additional materials (a file with the originals of blots, gels, membranes).

Response 2: The images included in the main text are sections of one of the membranes that showed the most clearly defined colonies. However, the reviewer’s point is indeed important. Therefore, we have added full membrane images corresponding to the stereomicroscope observations as supplementary material. Additionally, the image of group 1 at 56 days post-immunization (Figure 4, page 9) was replaced with one from a duplicate membrane that contained a greater number of colonies, although they were not as visually good.

Comment 3: How do the authors explain the difference in IFN-γ gene expression when stimulated by recombinant proteins? There, depending on the protein (and its concentration), the opposite effect is obtained on day 168 (Figure 6 A, B, 1 μg or 2 μg of the protein).

Response 3: We appreciate the reviewer’s insightful observation. This aspect may be related to the immunogenic characteristics of the proteins; however, it would still be necessary to evaluate other mechanisms and cytokines to understand how the response is coordinated by each one. Nevertheless, at 168 days, in a similarly closely associated manner, the gene expression of both cytokines, IFN-γ and IL-12, was increased with 1 µg of P40 and with 2 µg of MAG_1560. A late IFN-γ production response by antigen-stimulated cells may also occur as part of the immune response development.

Comment 4: What strain of M. agalactiae was used in the work? This should be stated in the materials and methods, not just in the legend to Figure 4.

Response 4: We thank the reviewer for the valuable comment. This point is indeed important; The M. agalactiae strain used was GM 139 and it is addressed in line 115 of the manuscript.

Comment 5: In Figure 2, at a quick glance, the identical letter designations of the groups of vaccinated animals and parts of the drawing corresponding to different vaccination options are confusing. Perhaps some things should be renamed; for example, renaming the groups 1, 2, 3 in the figure and in the corresponding places in the manuscript? At the discretion of the authors.

Response 5: We agree with the reviewer’s comment, as the suggested changes improve the clarity of the results and facilitate a better understanding. Accordingly, the suggested changes were made: group A was changed to group 1, group B to group 2, and group C to group 3. These changes can be observed in the following lines: 25, 26, 27, 32, 33, 34, 35, 153, 154, 176, 187, 200, 254, 258, 259, 262, 267, 273, 275, 276, 285, 287, 289, 290, 295, 306, 309, 310, 313, 316, 317, 325, 326, 329, 334, 337, 338, 345, 347, 352, 356, 359, 360, 368, 371, 376, 379, 380, 401, 407, 408, 473, 486, 499 and 516. And the changes were made in all the figures from the page 8 to 13.

4. Response to Comments on the Quality of English Language

Not applicable.

5. Additional clarifications

Based on the positive results obtained in this study, new immunization trials using the recombinant proteins are already being conducted to better understand the immune response and assess the vaccine's efficacy.

Reviewer 3 Report

Comments and Suggestions for Authors

The manuscript entitled "Immune responses induced by recombinant membrane proteins of Mycoplasma agalactiae in goats" addresses an important topic in veterinary immunology. The paper focused on the development of a recombinant subunit vaccine against the mention mycoplasma which is relevant given the economic impact of contagious agalactia and the limitations of current vaccination strategies. The study aims to evaluate the immunogenicity of specific recombinant proteins (P40 and MAG_1560) in goats to develop them as potential 87 antigens for formulating a recombinant subunit vaccine against contagious agalactia. The choice of these antigens is well-justified, and the authors demonstrated experience in the field. The paper is well-written. The experimental approach includes protein expression, immunization, and immune response assessment via ELISA and gene expression analysis, providing a comprehensive investigation. Also, the inclusion of both humoral and cellular immune response analyses enhances the depth of the immunogenicity assessment.

Specific comments:

L27, 100 and 120: Tris-buffered saline instead of tris-buffered saline

L77: MAG_6130 instead of MAG_ 6130

L92 and 241: kDa instead of KDa

L94: Which vector?

L98: β-D-1-thiogalactopyranoside instead of β-d-1-thiogalactopyranoside

L114 and 119: Add the speed of the centrifugation and the time that you centrifuged.

L139: in Tris-buffer of water?

L148: ad libitum instead of ad libitum

L179 and 192: did you use serum from group C as a negative control? Specify

L198: in vitro instead of in vitro

L203, 292, 427 and 456: in vitro instead of in vitro

L269: In Figure 3 A and B is it sodium thiocyanate or ammonium thiocyanate?

Discussion

The choice of Freund’s adjuvant should be justified, considering its known reactogenicity and its relevance to vaccine development. Why did you use the complete adjuvant in the first immunization and the complete one in the second. Please specify in the discussion.

Author Response

We sincerely thank you for the valuable suggestions regarding the writing. All recommended changes have been made, as shown below.

Comment 1: L27, 100 and 120: Tris-buffered saline instead of tris-buffered saline
Response 1:
The revision is shown in L27, L102, L123 and L156.

Comment 2: L77: MAG_6130 instead of MAG_ 6130
Response 2:
The revision is shown in L77.

Comment 3: L92 and 241: kDa instead of KDa
Response 3:
The revision is shown in L93 and L251.

Comment 4: L94: Which vector?
Response 4: pET-28A (+).
The revision is shown in L95.

Comment 5: L98: β-D-1-thiogalactopyranoside instead of β-d-1-thiogalactopyranoside
Response 5:
The revision is shown in L100.

Comment 6: L114 and 119: Add the speed of the centrifugation and the time that you centrifuged.
Response 6:
The revision is shown in L116 and L122.

Comment 7: L139: in Tris-buffer of water?
Response 7:
The revision is shown in L142.

Comment 8: L148: ad libitum instead of ad libitum
Response 8:
The revision is shown in L152.

Comment 9: L179 and 192: did you use serum from group C as a negative control? Specify
Response 9:
The revision is shown in L177. For avidity test the negative control was just PBS alone as indicated in L193.

Comment 10: L198: in vitro instead of in vitro
Response 10:
The revision is shown in L207.

Comment 11: L203, 292, 427 and 456: in vitro instead of in vitro
Response 11:
The revision is shown in L212, L304, L444 and L477.

Comment 12: L269: In Figure 3 A and B is it sodium thiocyanate or ammonium thiocyanate?
Response 12: Thank you for pointing this out. We have made the change in the image Figure 3 (page 8) to ammonium thiocyanate.

Comment 13: The choice of Freund’s adjuvant should be justified, considering its known reactogenicity and its relevance to vaccine development. Why did you use the complete adjuvant in the first immunization and the complete one in the second. Please specify in the discussion.
Response 13:
We appreciate the reviewer’s insightful observation. Freund’s adjuvant is still widely used in animal research to enhance the immune response. Since we are working with recombinant subunit antigens, we considered its use to be a suitable option for initial trials. As commonly practiced, complete Freund’s adjuvant was used for the first immunization to boost the response, while the incomplete form was used for the second dose in order to minimize adverse reactions. However, we emphasize that the use of this adjuvant was limited to this preliminary test. In future studies, we intend to evaluate different adjuvants that are more appropriate for formulations with commercial potential, but that are also capable of effectively stimulating and enhancing the immune response. We added a discussion regarding the use of Freund’s adjuvant in line 426.

4. Response to Comments on the Quality of English Language

Not applicable.

5. Additional clarifications

Based on the positive results obtained in this study, new immunization trials using the recombinant proteins are already being conducted to better understand the immune response, assess the vaccine's efficacy and formulation with other adjuvants.
